# Discrimination of Benign and Malignant Lesions in Canine Mammary Tissue Samples Using Raman Spectroscopy: A Pilot Study

**DOI:** 10.3390/ani10091652

**Published:** 2020-09-14

**Authors:** Diana Dantas, Liliana Soares, Susana Novais, Rui Vilarinho, J. Agostinho Moreira, Susana Silva, Orlando Frazão, Teresa Oliveira, Nuno Leal, Pedro Faísca, Joana Reis

**Affiliations:** 1IFIMUP, Department of Physics and Astronomy, Faculty of Sciences of University of Porto, Rua do Campo Alegre 687, 4169-007 Porto, Portugal; dddianadantas@gmail.com (D.D.); soareslilianasantos@gmail.com (L.S.); rvsilva@fc.up.pt (R.V.); jamoreir@fc.up.pt (J.A.M.); 2INESC TEC—Institute for Systems and Computer Engineering, Technology and Science, Rua do Campo Alegre 687, 4169-007 Porto, Portugal; susana.novais@inesctec.pt (S.N.); susana.o.silva@inesctec.pt (S.S.); orlando.frazao@inesctec.pt (O.F.); 3Departamento de Medicina Veterinária; Escola de Ciências e Tecnologia; Universidade de Évora; Pólo da Mitra, Apartado 94, 7002-554 Évora, Portugal; teresoliveira@uevora.pt; 4DNAtech Laboratório Veterinário, Estrada do Paço do Lumiar, N.° 22 Edifício E, 1° Andar, 1649-038 Lisboa, Portugal; lealnunom@gmail.com (N.L.); pedrofaisca@ulusofona.pt (P.F.); 5Centro de Investigação em BioCiências e Tecnologias da Saúde, Faculdade de Medicina Veterinária, Universidade Lusófona de Humanidades e Tecnologias, Campo Grande 376, 1749-024 Lisboa, Portugal

**Keywords:** breast cancer, histopathology, microcalcification, canine, comparative oncology, Raman spectroscopy, Raman fingerprints

## Abstract

**Simple Summary:**

Neoplastic diseases are among the leading causes of death worldwide and constitute the main health problem in both human and veterinary medicine, particularly as the occurrence of the disease continues to increase. Comparative oncology is a quickly expanding field that examines both cancer risk and tumor development across species. Characterized by interdisciplinary collaboration, its goal is the improvement of both human and animal health. Canine neoplastic disease occurs spontaneously and has comparable clinical presentation and pathophysiology to corresponding human cancers. Since the nature of the disease is spontaneous, the complex interactions between tumor cells, tissues and the immune system can be better studied. Such relations are otherwise difficult to study in other experimental animal models. Raman spectroscopy has proved to be a suitable technique to detect and study breast microcalcifications. Raman spectroscopy is a specific and sensitive tool for identifying biomarkers of oncologic disease and also shows further potential in differentiating malignant and benign tumors, and these tumors from healthy tissue.

**Abstract:**

Breast cancer is a health problem that affects individual life quality and the family system. It is the most frequent type of cancer in women, but men are also affected. As an integrative approach, comparative oncology offers an opportunity to learn more about natural cancers in different species. Methods based on Raman spectroscopy have shown significant potential in the study of the human breast through the fingerprinting of biological tissue, which provides valuable information that can be used to identify, characterize and discriminate structures in breast tissue, in both healthy and carcinogenic environments. One of the most important applications of Raman spectroscopy in medical diagnosis is the characterization of microcalcifications, which are highly important diagnostic indicators of breast tissue diseases. Raman spectroscopy has been used to analyze the chemical composition of microcalcifications. These occur in benign and malignant lesions in the human breast, and Raman helps to discriminate microcalcifications as type I and type II according to their composition. This paper demonstrates the recent progress in understanding how this vibrational technique can discriminate through the fingerprint regions of lesions in unstained histology sections from canine mammary glands.

## 1. Introduction

Breast cancer is the most common type of cancer among women, and it has an impact on 2.1 million women per year. This disease causes the highest number of cancer-related deaths among women. In 2018, the World Health Organization (WHO) estimated that 627,000 women died of breast cancer—that is, about 15% of all cancer deaths among women. Although breast cancer rates are higher among women in more developed regions, rates are increasing in almost all regions globally [1].

Carcinogenesis is a multistage process in which cancer develops. In this process, several abnormal cellular changes occur that can generate metastases, which promotes neoplasm growth. Tumors may be classified as benign or malignant based on histopathology [2]. In addition, breast tissue microcalcifications are often used as markers for the type of breast pathology and can act as strong differentiators between benign and malignant, proliferative breast lesions (type I and type II, respectively) and as indicators of the presence of early breast cancer [3].

The occurrence of spontaneous tumors in certain animals makes them suitable models for the study of tumorigenesis, for improvement of diagnostic tools and for the development of successful therapies in humans. In addition, the shorter lifespan combined with a faster progression of cancer in model species allows for the completion of trials and faster data collection [4]. Experimental animal models of carcinogenesis have been used as models for human breast cancer and Raman spectroscopy has been used to assess the aggressiveness of tumors [5]. Spontaneous mammary tumors are common in dogs and cats and about 42% of all tumors in female dogs and 17% in cats affect the mammary gland [6,7].

Unlike the usual laboratory animals such as rodents, cats and dogs share the same environmental risk factors as humans. Thus, cancer occurs naturally in this species, unlike rodents, which are often immunocompromised [8]. Dogs have an incidence of breast cancer about three times higher than in women [9]. Dogs develop natural cancers whose biology bears several strong similarities with cancer in humans, including patterns of response/resistance to conventional therapy, metastases, and recurrence [10,11]. As with humans, microcalcifications are present and morphologically significantly differ in non-neoplastic canine mammary tissue, and benign and malignant mammary tumors, further validating this recognized spontaneous animal model and microcalcifications as biomarkers of mammary cancer [12,13].

In recent decades, Raman spectroscopy has received considerable attention with regard to its clinical oncology applications, for ex vivo studies and during surgery of neoplasms affecting the brain, breast, lung, skin, uterus, and GI tract [14,15,16,17]. In fact, our research group has successfully shown the feasibility of Raman spectroscopy for discriminating between adenocarcinomatous and normal mucosal formalin-fixed colonic tissues [18]. Due to the major impact of breast cancer in women all over the world, Raman spectroscopy has been used extensively to investigate breast tissue and it is as a promising technique in breast cancer diagnosis [1,19]. The principle of it is based on the dispersion of inelastic light, also known as Raman dispersion, which generates biochemical fingerprints. These fingerprints reflect the biological composition and activity of tissues. Several groups have demonstrated that healthy and malignant breast tissue produces distinct Raman spectra [20]. Thus, Raman spectroscopy can provide a biochemical fingerprint of a tissue biopsy, and together with advanced data analysis techniques, has been shown to classify normal, benign, pre-cancer, and cancer cases [21]. This technique has emerged as a non-destructive analytical tool for the biochemical characterization of tissues, presenting numerous advantages, such as sensitivity to small structural changes, noninvasive sample capacity, and non-necessary sample preparation [1,22].

To the authors’ knowledge, Raman has also been used previously in canine mammary neoplasms by Birtoiu et al. [6] and in two clinical cases [23]. Both works used ex vivo, freshly excised, unprocessed samples, and a 632 nm wavelength laser. Birtoiu et al. refer to ten samples from female cats and dogs, without specification and the histopathological diagnosis was not described. Munteanu et al. refer to a case of solid carcinoma and another with lobular hyperplasia.

The present study aimed to explore the use of Raman spectroscopy to characterize, in histological sections, the lesions associated with different neoplasms in canine mammary tissue, with a focus on those with mineralization, and explore its potential to discriminate between benign and malignant lesions in this species, and systematically identifying characteristic fingerprints through a comparative oncology approach.

## 2. Materials and Methods

This research aimed to provide Raman spectra of mammary gland tissue in the neoplastic condition with benign and malignant lesions, and to also characterize the calcifications present.

This study used samples obtained by routine mastectomy, which is the standard treatment for canine mammary tumors. Mammary glands from six female dogs were excised by regional or complete mastectomy and fixed in 10% buffered formalin. A randomized alphanumeric code identified tissues. After histopathology, the remainders of the tissues are usually destined for destruction. Samples in this study were collected from those remaining tissues under fluoroscopic guidance for the detection of calcified areas. This study was authorized by the ORBEA-UÉvora.

### 2.1. Canine Mammary Samples

Canine mammary samples from six female dogs were used in this study. Under fluoroscopy, seven biopsies were taken from areas with increased radiopacity. The biopsies were cut into thin longitudinal sections, two to three per biopsy, decalcified with RDO^®^ Rapid Decalcifier as necessary for sectioning, routinely processed for paraffin embedding, sectioned into 3 µm slides for histopathology and 8 µm for Raman spectroscopy. Sequential sections from these biopsies were used for histopathology analysis and Raman spectroscopy. For histopathology, 3 µm sections were stained with hematoxylin-eosin (H&E) and assessed by a veterinary pathologist. Mammary tumors were classified according to the Goldschmidt et al. proposal [24] and graded according to an adaptation of the Elston and Ellis grading system developed by Peña et al. for canine mammary tumors [25]. Differential staining with the Von Kossa-Van Gieson modified method was also performed to confirm the presence of mineralization. For Raman spectroscopy, sections were routinely deparaffinized and hydrated. Table 1 summarizes the information on the canine mammary samples and histopathology, illustrated in Figure 1.

### 2.2. Raman Spectroscopic Measurements

Raman spectroscopy was performed using a Renishaw InViaTM Qontor^®^ Confocal spectrometer, equipped with lasers of different wavelengths of 532, 633 and 785 nm. The 532 nm laser line was chosen for excitation of all samples under analysis as the remaining wavelengths provided strong background fluorescence. The samples were hydrated and placed in the microscope platinum. A 50× lens was used for focusing the laser beam on the sample. The incident power on the sample surface was kept at 3.3 mW in order to avoid sample damage. Each Raman spectrum was acquired with an exposure time of 10 s. The scattered light was analyzed by using a 1200 lines/mm diffraction grating, providing a spectral resolution of 2 cm^−1^. The 521 cm^−1^ band of Si was used for calibration. The spectra were recorded from different sample regions containing clinically significant morphological changes, which had been previously marked by a veterinary pathologist. In order to obtain information regarding tissue heterogeneity, several spectra were obtained. Considering the consistency of spectra at each given point and the heterogeneous nature of canine mammary tumors, a minimum of three spectra per site was acquired, from at least 15 sites per sample. Table 2 summarizes the number of sites examined in each section and the number of spectra obtained under the same experimental conditions.

### 2.3. Spectral Preprocessing

In order to correct for overall intensity and background fluctuations in the entire data set, the normalization criteria that was adopted comprises background subtraction and intensity correction. The background was estimated using data analysis software by fitting a second order polynomial to each spectrum, as is common practice for tissue fluorescence elimination. The obtained data were subsequently smoothed using a Fast Fourier Transform filter (FFT filter). Thus, the quality of the processed Raman data was improved, which contributed to proper data analysis.

## 3. Results and Discussion

In this section, the experimental results of the Raman studies on the neoplastic tissues and calcifications of the canine mammary tissues are presented.

All tissue samples were previously tested by the three lasers available in the Raman spectrometer system, i.e., emitting at the wavelengths of 532, 633 and 785 nm. The Raman spectra acquired with 633 and 785 nm laser lines shown strong background fluorescence, inhibiting the analysis of the acquired data. The Raman spectra acquired with the 532 nm laser presented much lower fluorescence, and thus, it was chosen for the excitation of all samples for further analysis, an approach applied by other authors [16,21,22,26,27,28]. Even so, it was impossible to obtain clear Raman signals in three out of the seven biopsy sections, due to laser-induced fluorescence. This is a common and serious limitation when using Raman spectroscopy in microbial, plant, and animal cells [26]. Autofluorescence can be produced by tryptophan and tyrosine side chains in folded proteins, by carotenoids and photosynthetic pigments or other chromophores [29]. Incomplete removal of paraffin may also generate fluorescence and interfere with the Raman signal [30]. Several approaches are possible to circumvent these limitations.

Only one of two previous published studies on canine mammary neoplasms [6,23] refers to the histopathological diagnosis of two clinical cases [23], which were both different from the ones included in the present study and none showed mineralization. These studies were performed in freshly excised tissue, unlike our samples, and Raman spectra were acquired with a 632 nm laser, unlike the present study.

Concerning the analysis of the acquired data, two regions in the Raman spectra are of potential interest for breast cancer diagnostics: The Raman fingerprint (FP) region (400–1800 cm^−1^) and the high wavenumber (HW) region (2800–3200 cm^−1^), where the Raman bands assigned to the lipid/protein vibrations are expected [20]. The former spectral regions exhibit Raman bands assigned to the vibration of different biomolecules, which were mostly lipids, proteins, carotenoids, amino acids, and nucleic acids. Vibrations of microcalcifications can be found in this spectral range when they are present in the samples.

In the present work, only data obtained using the same methodology sample preparation and Raman spectroscopy acquisition were included and concerned biopsies #1, #2-1, #3, and #4, all from benign and malignant tumors (see Table 2).

Three of the biopsies corresponded to mixed-type tumors (samples #1, #2-1, #3) with the presence of extensive bone metaplasia. Mixed tumors are the most common mammary tumors in dogs, while metaplastic breast cancer is rare in humans. However, metaplastic breast cancer and canine mixed tumors share histopathologic characteristics and molecular expression profiles [31]. The Raman peaks common to mineralized areas, and found in these samples are summarized in Table 3.

The following Section 3.1 and Section 3.2 present the Raman spectroscopy results of samples #1 and #3, respectively, and the analysis of the major fingerprints obtained with the proposed technique, namely, the ones presented in Table 3. Due to similar results achieved from the analysis of Raman spectra of samples #1 and #2, apart from being two mixed-type benign tumors with bone metaplasia, only the results from Sample #1 are shown. Section 3.3 briefly discusses the major Raman assignments achieved from the analysis of sample #4—a malignant tumor. Although it does not have bone metaplasia, it presents Raman peaks in common with samples #1 to #3, as they all share tissue changes related with neoplasia. Section 3.4 summarizes the main Raman fingerprints achieved for all samples under study (#1 to #4) and its major assignments.

### 3.1. Raman Spectroscopy of Sample #1

Figure 2 shows the spectra for the sample #1 study. The Raman spectra show peaks that the literature indicates are suggestive of neoplastic changes and mineralization. The inset in Figure 2 is a 2.5× microscopic image of a section of sample #1 where mineralized areas are stained black by means of the Von Kossa-Van Gieson modified method. A 50× microscopic image is also depicted on the right side of Figure 2, representing a mineralized area of sample #1.

In the spectral range of 400–1800 cm^−1^, it was verified that the areas identified with metaplasia showed peaks that indicate their neoplastic nature. The bands at 428 and 589 cm^−1^ are assigned to phosphate vibrations [26]. Also, there is a well-defined phosphate band at 960 cm^−1^ and a much weaker carbonate band at 1070 cm^−1^, both attributed to calcifications called type II, composed of calcium hydroxyapatite (HA). HA deposits occur in both benign and malignant tumors; HA is also a normal bone component, so it would be expected to be present in association with bone metaplasia [31,33]. However, the effects of demineralization during previous sample processing on the calcified areas must be carefully considered. The remaining peaks have a contribution mainly from protein and are associated with nucleic acids, which are characteristic of cancerous tissues [2]. In particular, the band at 1673 cm^−1^ is assigned to the amide I mode originating mainly from the polypeptide backbone and protein secondary structure. This Raman band has also been related to nucleic acid modes and was identified as indicative of tissue changes associated with disease, namely, neoplasia [20,34,35]. In the spectral range 2800–3000 cm^−1^, the bands located at 2883 and 2938 cm^−1^ are assigned to the protein contribution associated with cancerous tissue [36]. The main band of proteins that is typically at 2939–2944 cm^−1^ is due to the aromatic and aliphatic amino acids, the charged amino acids and proline, threonine, and histidine [29]. Table 4 details the position of the peaks analyzed and compares these with the peaks reported in the literature, and describes the mode signatures present in the areas of metaplasia.

The second sample region analyzed presents fingerprints that are distinct from the first. Figure 3 shows the spectra of the analyzed areas containing glandular secretions and cellular debris inside an ectasia duct in sample #1, and the inset shows a microscopic image (2.5×) of said areas stained by means of the Von Kossa-Van Gieson modified method. On the right side of Figure 3 there is a microscopic image (50×) of the analysed areas in sample #1 containing glandular secretions and cellular debris.

The peaks 1096 and 1295 cm^−1^ do not show a protein contribution, as they are assigned to phosphodioxy (${\mathrm{PO}}_{2}^{-}$) groups and methylene twisting, respectively [34]. The 1440 cm^−1^ peak is an indicator of lipids, a Raman biomarker for healthy tissue [22,36]. In the high wavenumber spectral region, there are four bands, unlike the region previously analyzed. These bands correspond to lipid biomarkers not associated with tumors and present in healthy tissues.

Table 5 shows in detail the position of the analyzed peaks and the comparison with the peaks reported in the literature, and describes the mode signatures present in the secretions area. The peak 558 cm^−1^ is mentioned in the literature as being associated with silica [39]. Considering the area where it was identified, it was most likely generated by the glass slide that held the section, rather than by mineralization associated with bone metaplasia.

Figure 4 shows the Raman spectra of the demineralized regions in the benign mixed tumor and, the inset shows a microscopic image (2.5×) of said areas stained magenta by means of the Von Kossa-Van Gieson modified method. On the right side of Figure 4 there is a microscopic image (50×) of a demineralized region in sample #1.

This area, unlike the previous region, reveals peaks that indicate it is not healthy. There are peaks associated with nucleic acids, 785 and 1455 cm^−1^, and Amides III and I, 1250 and 1673 cm^−1^, respectively, which are potential indicators of injury [34,39]. Peaks at 2883 and 2942 cm^−1^, such as in areas of metaplasia, correspond to a vibrational mode associated with proteins that can contribute to the neoplastic character [34].

The analysis of this tissue allows us to infer that the results match the histopathology, and the tissue shows mineralization points. Table 6 shows the position of the peaks analyzed and the comparison with the peaks reported in the literature, and describes the mode signatures present in breast parenchyma.

### 3.2. Raman Spectroscopy of Sample #3

We also analyzed sample #3, which was characterized histologically as mixed carcinoma.

Initially, it is possible to see that it is similar to the fingerprint of the benign sample. However, there are significant differences in the wavenumber of some Raman peaks, which reinforce the structural protein modes’ indicators.

Figure 5 and Table 7 show the presence of the same chemical compounds, again highlighting the presence of the type II calcifications and HA corresponding to the phosphate peak at 960 cm^−1^ is very prominent, with the less intense carbonate peak at 1070 cm^−1^.

By analyzing these spectra, it is possible to observe its similarities with the breast parenchyma that was previously analyzed (Figure 2). Table 7 shows the analysis of the peaks of tissue in the metaplastic areas.

However, there are clear differences in the non-metaplastic mammary tissue areas (Figure 6 and Table 8). There is a remarkable difference in the peak of 1450 cm^−1^. The presence of this peak indicates a CH_2_ bending mode in malignant tissues [32]. In addition, there is a nucleic acid and protein contribution from the other peaks obtained, 785, 1322, 1673, 2882, and 2937 cm^−1^, which confirms the identification of a malignant neoplasm. The increasing contribution of protein components is a well-known indicator of carcinogenesis [5,27,32,34].

From the results presented so far, it can be seen that the samples have similar fingerprints and the same type of calcifications when compared by region. However, there are differences in the wavenumbers that arise from different neoplastic conditions, dependent on malignancy.

The ratio of phosphate and carbonate Raman bands at 960 and 1070 cm^−1^ helps to determine the carbonate content. Benign lesions have more calcium carbonate and fewer proteins compared to malignant lesions [32,41]. However, some of the present samples have been partially decalcified to allow sectioning. Ongoing work is being developed in formalin-fixed specimens in order to characterize mineralization in canine benign and malignant tumors to explore whether the morphological differences previously described [12,13] translate, as in humans, into different chemical composition and crystallite growth, and therefore, differences in the correspondent Raman spectra.

### 3.3. Raman Spectroscopy of Sample #4

Sample #4 produced very heterogenous Raman spectra throughout the section surface, and translated the variety of changes observed in the histopathologic assessment of this highly aggressive tumor. However, all generated spectra in this sample showed markers of malignancy such as the Raman peaks related to proteins, collagen, DNA, and nucleic acids—this is a feature of malignancy as it translates into protein-dominated spectra. In contrast, healthy tissues show several Raman bands assigned to lipids.

This was the only sample showing the band at 1232 cm^−1^, assigned to amide III, a protein band that is closely linked to breast cancer [20,27,35], although sample #1 shows a 1250 cm^−1^ peak, associated with vibrational phosphate, possibly associated with amide III [40].

### 3.4. Main Raman Fingerprints

Table 9 summarizes the differentiating peaks for the analyzed sections, including those that have previously been considered as malignancy fingerprints.

The band at 1440 cm^−1^ is an indicator of lipids, a Raman biomarker for healthy tissue [22,28]. The band at 1665 cm^−1^ of amide I is a collagen assignment [20,34,35].

The band at 1322 cm^−1^ corresponds to CH_3_/CH_2_ twisting and wagging in collagen, which is found in malignant tissues due to the change in the molecular structures of proteins associated with tumor transformations [42].

The band at 1450 cm^−1^ is generally attributed to the CH_2_ bending mode in malignant tissues and seems to have a contribution from albumin. This is understandable since it is known that increased blood flow in the microcirculation at the site of injury is present in inflammatory conditions, and increased inflammatory changes associated with mammary tumors have been associated with poorer prognosis in dogs [43,45].

As previously discussed, the band at 1673 cm^−1^ is assigned to the amide I mode originating mainly from the polypeptide backbone and protein secondary structure and it is present in all samples (#1 to #4). This Raman band has been identified as indicative of tissue changes associated with disease, namely, neoplasia [20,34,35].

It is also worth noting that all samples (#1 to #4) have the Raman band at 1454 cm^−1^ in common, which is assigned to CH_3_ and CH_2_ deformation (proteins), an indicator of unhealthy tissues [34]. Tissue samples #1 and #2 also have the band at 1452 cm^−1^ in common, which is assigned to the C-H bending mode and it has been associated with lipids and cardiac valvular calcification [27,37]; 1453 cm^−1^ has been associated with the structural modes of proteins in tumors [34].

The main difficulty in interpreting the Raman and Imaging Raman bands in the region of 2800–3000 cm^−1^ results from the fact that the spectral region of the fatty and lipid acid bands overlaps the protein bands. However, challenges are also posed by the heterogeneous nature of canine mammary tumors that makes laser focusing in a uniform cell population unlikely. The existence of strong laser-induced fluorescence in the non-neoplastic samples included initially also limits the comparison with non-neoplastic tissues with the present methodology of sample preparation. The research team is currently working on solutions to circumvent autofluorescence, allowing acquisition from control tissues, application of the methodology to a larger number of samples, and better representative of the diversity of canine mammary tumors with subsequent statistical analysis.

## 4. Conclusions

This study has illustrated important features of Raman spectroscopy reports that may be used to characterize mammary tissue at a biochemical, molecular level. A thorough Raman spectral reading enabled the identification of fingerprints consistent with tumoral lesions and mineralized areas, as identified in the histopathology. A detailed analysis was done in the FP (400–1800 cm^−1^) and HW (2800–3200 cm^−1^) regions. In the FP region, the band 960 cm^−1^ associated with type II calcification is prominent and the most intense since three of the samples had mineralized bone metaplastic regions, as confirmed by Von Kossa staining. Furthermore, the peak 1450 cm^−1^ associated with the CH_2_ bending mode and consistent with structural protein changes, was previously described as part of the fingerprint for breast lesions malignancy and this was present in the mixed carcinoma sample, but not in the benign mixed tumor samples. The carcinosarcoma presented a marked predominance of peaks associated with proteins and nucleic acids, and the spectral heterogenicity was consistent with the malignant changes affecting the epithelium, the myoepithelial elements and atypical cartilage and bone. The HW region will be subject to further study due to the spectral degradation observed from benign to malignant tissues. This behavior is somehow an indicator of malignancy as the peaks found are associated with proteins that, in turn, are increased in neoplastic conditions. The behavior of phosphate and carbonate peaks at 960 and 1070 cm^−1^, respectively, will also be subject to further analysis, in terms of evaluating the relative amount of carbonate substitution and degree of crystallinity—as a potential method of distinguishing between type II calcifications, either being associated to benign or malignant tissues.

These findings were consistent throughout the samples and supported the use of Raman spectroscopy for complementary clinical application, in diagnosis and in helping to define surgical margins since some of the obstacles found arise from previous sample processing. The results, although preliminary, further support the use of canine spontaneous mammary tumors as a model for human breast cancer while reinforcing the potential of Raman spectroscopy for identification of biomarkers of animal disease, in parallel to its use in humans.

## Figures and Tables

**Figure 1 animals-10-01652-f001:**
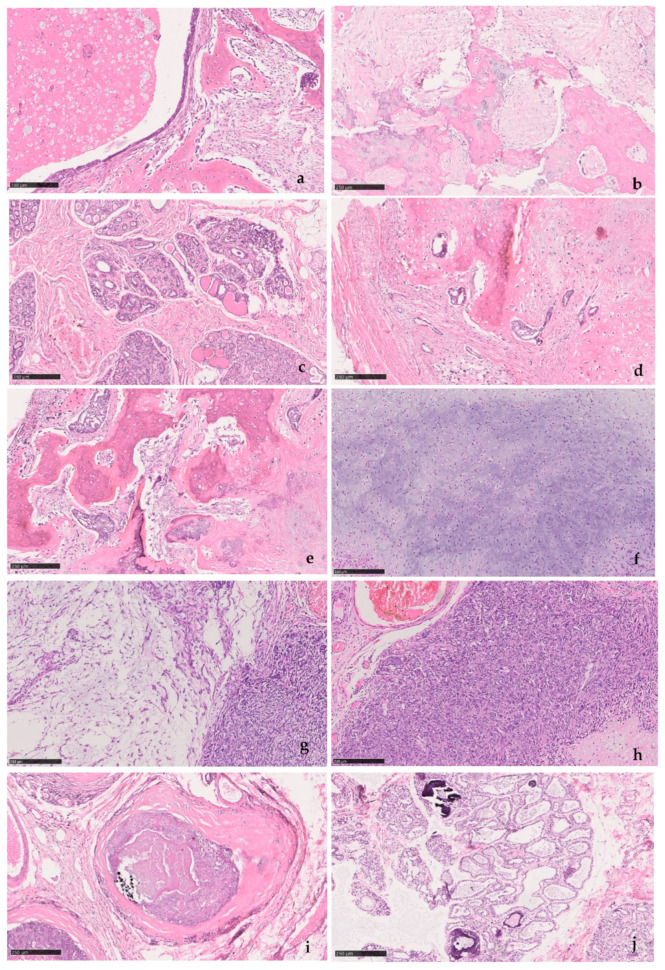
Histopathology findings (H&E): (**a**) #1 Mixed benign tumor; (**b**) #2-1 Mixed benign tumor; (**c**) #2-2 Glandular tissue; (**d**,**e**) #3 Mixed malignant tumor, with epithelial and mesenchymal cell proliferation and foci of bone and chondroid metaplasia; (**f**) #4 Carcinosarcoma (mesenchymal); (**g**) #4 Carcinosarcoma (myoepithelial and epithelial); (**h**) #4 Carcinosarcoma (epithelial); (**i**) #5 Comedocarcinoma; (**j**) #6 Glandular tissue with mineralization foci within acinar ducts.

**Figure 2 animals-10-01652-f002:**
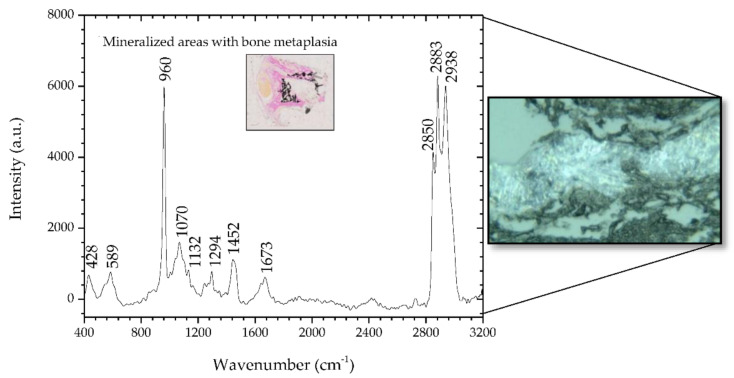
Raman spectra of mineralized areas of a benign mixed tumor (sample #1) and, on the right side, the corresponding microscopic image (50×, Raman spectrometer system). The inset shows the microscopic image (2.5×) of a section of sample #1 where mineralized areas are stained black by means of the Von Kossa-Van Gieson modified method.

**Figure 3 animals-10-01652-f003:**
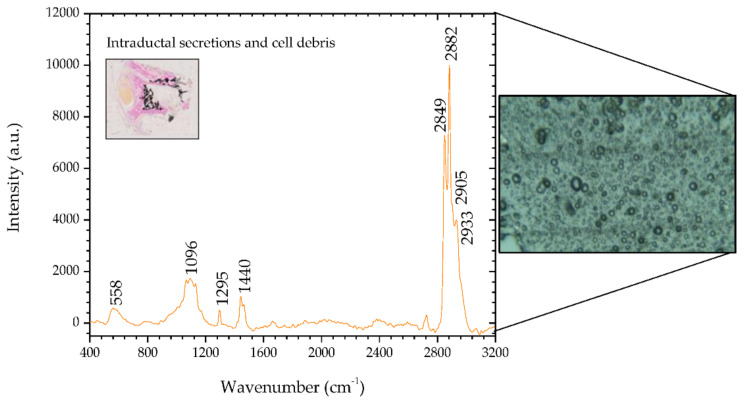
Raman spectra of intraductal secretions and cellular debris in ectasia duct areas in sample #1 and, on the right side, the corresponding microscopic image (50×, Raman spectrometer system). The inset shows a microscopic image (2.5×) of intraductal secretions and cellular debris stained orange by means of the Von Kossa-Van Gieson modified method.

**Figure 4 animals-10-01652-f004:**
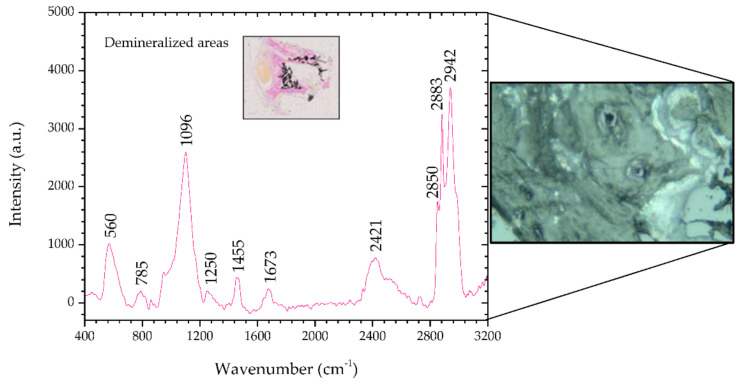
Raman spectra of demineralized areas of sample #1 and, on the right side, the corresponding microscopic image (50×, Raman spectrometer system). The inset shows the microscopic image (2.5×) of demineralized areas stained by means of Von Kossa-Van Gieson modified method.

**Figure 5 animals-10-01652-f005:**
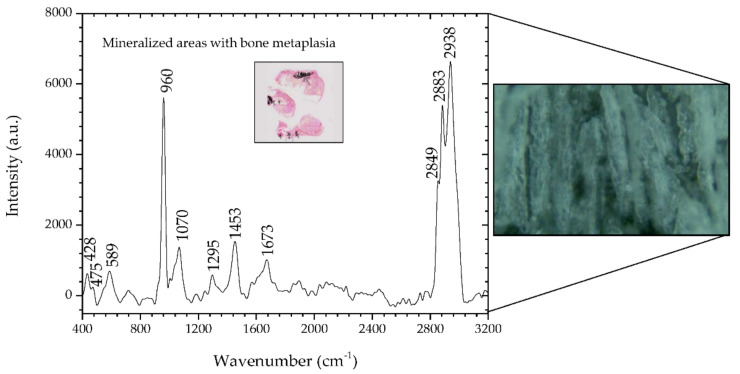
Raman spectra of mineralized areas of bone metaplasia of the mixed carcinoma (sample #3) and, on the right side, the corresponding microscopic image (50×, Raman spectrometer system). In the inset is the microscopic image (2.5×) of a section of sample #3 where mineralized areas are stained black by means of the Von Kossa-Van Gieson modified method.

**Figure 6 animals-10-01652-f006:**
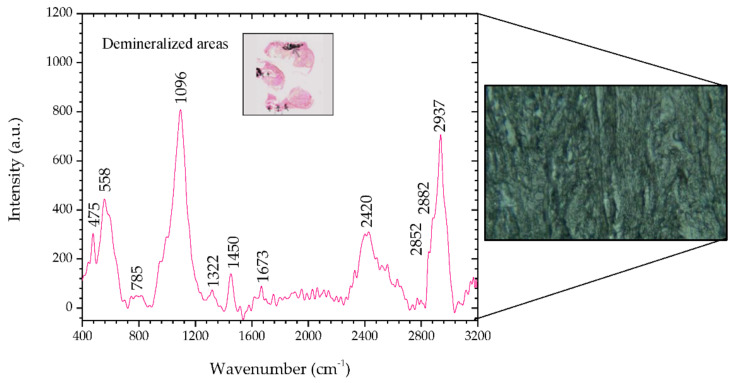
Raman spectra of demineralized areas of the mixed carcinoma (sample #3) and, on the right side, the corresponding microscopic image (50×, Raman spectrometer system). The inset shows a microscopic image (2.5×) of demineralized areas stained by means of Von Kossa-Van Gieson modified method.

**Table 1 animals-10-01652-t001:** Identification of the samples under study and results of histopathological analysis.

Individual	Biopsies	Age (Years)	Type	Histopathology Biopsy Findings
#1	1	10	Benign	Mixed benign tumour
				Foci of bony and cartilaginous metaplasia
#2	1	7	Benign	Mixed benign tumour
				Foci of bony and cartilaginous metaplasia
	2		Non-neoplastic	Glandular tissue
#3	1	10	Malignant	Mixed carcinoma
				Foci of chondroid metaplasia and epithelial cells
#4	1	11	Malignant	Carcinosarcoma
#5	1	14	Malignant	Comedocarcinoma
#6	1	10	Non-neoplastic	Glandular tissue

**Table 2 animals-10-01652-t002:** Number of sites and number of spectra per site obtained in each sample.

Individual	Biopsies	Type	Raman Spectroscopy Acquisition
#1	1	Benign mixed tumor	Mineralized area—10 sites, 3 spectra each
			Demineralized area—9 sites, 3 spectra each
			Secretions inside ectasia duct—6 sites, 3 spectra each
#2	1	Benign mixed tumor	Calcification type II—12 sites, 3 spectra each
			Parenchyma—9 sites, 3 spectra each
	2	Non-neoplastic glandular tissue	Strong background fluorescence throughout the sample
#3	1	Mixed carcinoma	Mineralized are—11 sites, 3 spectra each
			Parenchyma—15 sites, 3 spectra each
#4	1	Carcinosarcoma	15 sites, 3 spectra each
#5	1	Comedocarcinoma	Strong background fluorescence throughout the sample
#6	1	Non-neoplastic glandular tissue	Strong background fluorescence throughout the sample

**Table 3 animals-10-01652-t003:** Peaks common to the samples of mixed type tumors with bone metaplasia.

Individual	Type	Raman	
		Peaks (cm^−1^)	Major Assignment
#**1**	Benign	960	Phosphates—Microcalcification type II [24,29,32]
#**2-1**	Benign	1070	Carbonates—Microcalcification type II [24,29,32]
#**3**	Malignant	1673	Amide I (proteins) [26]

**Table 4 animals-10-01652-t004:** Peak position and mode assignment of mineralized areas of benign mixed tumor (sample #1).

Peaks (cm^−1^)		Major Assignment
This Work	Literature [27,34,36,37,38]	
428	428	Symmetric stretching vibration of ν_2_ ${\mathrm{PO}}_{4}^{3-}$ (phosphate of HA)
589	589	Symmetric stretching vibration of ν_4_ ${\mathrm{PO}}_{4}^{3-}$ (phosphate of HA)
960	960	Symmetric stretching vibration of ν_1_ ${\mathrm{PO}}_{4}^{3-}$ (phosphate of HA); Calcium-phosphate stretch band (high quantities of cholesterol); Quinoid ring in-plane deformation—**Microcalcification Type II**
1070	1070	Calcium-carbonate ν_1_ mode (carbonate of HA)
1132	1131	Palmitic acid; Fatty acid
1294	1294	Methylene twisting
1452	1452	Protein peaks; Umbrella mode of methoxyl; C-H bending mode of structural proteins and lipids; structural protein modes of tumors; associated with mineralization
	1453	
1673	1673	Amide I
2850	2850	ν_s_ CH_2_, lipids, fatty acids; CH_2_ symmetric
2883	2883	CH_2_ asymmetric stretch of lipids and proteins
2938	2940	C-H vibrations in lipids & proteins; ν_as_ CH_2_, lipids, fatty acids

**Table 5 animals-10-01652-t005:** Peak position and mode assignment of breast tissue of secretions area of benign mixed tumor (sample #1).

Peaks (cm^−1^)		Major Assignment
This Work	Literature [22,28,33,34]	
558	558	Silica dioxide bending modes
1096	1096	Phosphodioxy (${\mathrm{PO}}_{2}^{-}$) groups
1295	1294	Methylene twisting
	1296	CH_2_ deformation
1440	1440	CH_2_ and CH_3_ deformation vibrations; CH deformation: Cholesterol, fatty acid band: δ(CH_2_) (lipids); CH_2_ bending (lipids)
2849	2850	ν_s_ CH_2_, lipids, fatty acids; CH_2_ symmetric
2882	2883	CH_2_ asymmetric stretch of lipids and proteins
2905	2900	CH stretch
2933	2933	CH_2_ asymmetric stretch

**Table 6 animals-10-01652-t006:** Peak position and mode assignment of demineralized areas of benign mixed tumor (sample #1).

Peaks (cm^−1^)		Major Assignment
This Work	Literature [22,28,34,39,40]	
560	558, 646	Silica dioxide
785	785	U, T, C (ring breathing modes in the DNA/RNA bases); Backbone O-P-O
1096	1096	Phosphodioxy (${\mathrm{PO}}_{2}^{-}$) groups
1250	1250	Phosphate associated amide III
1455	1455	Deoxyribose; δ(CH_2_)
1673	1673	Amide I
2421	-	No reference found
2850	2850	ν_s_ CH_2_, lipids, fatty acids; CH_2_ symmetric
2883	2883	CH_2_ asymmetric stretch of lipids and proteins
2942	2940	C-H vibrations in lipids & proteins; ν_as_ CH_2_, lipids, fatty acids

**Table 7 animals-10-01652-t007:** Peak position and assignment of mineralized areas of bone metaplasia of the mixed carcinoma (sample #3).

Peaks (cm^−1^)		Major Assignment
This Work	Literature [34,38]	
428	428	Symmetric stretching vibration of ν_2_ ${\mathrm{PO}}_{4}^{3-}$ (phosphate of HA)
475	477	Polysaccharides, amylose, amylopectin
589	589	Symmetric stretching vibration of ν_4_ ${\mathrm{PO}}_{4}^{3-}$ (phosphate of HA)
**960**	960	Symmetric stretching vibration of ν_1_ ${\mathrm{PO}}_{4}^{3-}$ (phosphate of HA); Calcium-phosphate stretch band (high quantities of cholesterol); Quinoid ring in-plane deformation—**Calcification** **Type II**
1070	1070	Calcium-carbonate ν_1_ mode (carbonate of HA)
1295	1294	Methylene twisting
**1453**	1453	Protein peaks; Umbrella mode of methoxyl; C-H bending mode of structural proteins; **Structural protein modes of tumors**
1673	1673	Amide I
2849	2850	ν_s_ CH_2_, lipids, fatty acids; CH_2_ symmetric
2883	2883	CH_2_ asymmetric stretch of lipids and proteins
2938	2940	C-H vibrations in lipids & proteins; ν_as_ CH_2_, lipids, fatty acids

**Table 8 animals-10-01652-t008:** Peak position and assignment of demineralized areas of the mixed carcinoma (sample #3).

Peaks (cm^−1^)		Major Assignment
This Work	Literature [34,39]	
475	477	Polysaccharides, amylose, amylopectin
558	558	Silica dioxide
785	785	U, T, C (ring breathing modes in the DNA/RNA bases); Backbone O-P-O
1096	1096	$\mathrm{Phosphodioxy}\;({\mathrm{PO}}_{2}^{-}$) groups
1322	1322	CH_3_CH_2_ twisting, collagen; CH_3_/CH_2_ twisting and wagging in collagen; CH_3_CH_2_ deforming modes of collagen and nucleic acids
**1450**	1450	**CH_2_ bending mode in malignant tissues**; Bending modes of methyl groups (one of vibrational modes of collagen); Methylene deformation in biomolecules; **CH_2_ deformation in IDC breast tissue**; C-H deformation peaks (CH functional groups in lipids, amino acids side chains of the proteins and carbohydrates); δ(C-H); CH_2_ bending (proteins)
1673	1673	Amide I
2420	-	Not reference found
2852	2850	ν_s_ CH_2_, lipids, fatty acids; CH_2_ symmetric
2882	2883	CH_2_ asymmetric stretch of lipids and proteins
2937	2940	C-H vibrations in lipids & proteins; ν_as_ CH_2_, lipids, fatty acids

**Table 9 animals-10-01652-t009:** Differentiating peaks in the analyzed sections.

Individual	Raman	
	Peaks (cm^−1^)	Major Assignment
#1	1440	fatty acid band: δ(CH_2_) (lipids) [22,34]
#2	1665	Amide I (collagen assignment) [20,34,35]
#3	**1322**	**CH_3_/CH_2_ twisting (collagen)** [34,42]
	**1450**	**CH_2_ bending mode in malignant tissues** [25,34,43]
#4	**981**	**C-C stretching****β****-sheet (proteins)** [33,34,35]
	1002	Phenylalanine (collagen) [32,34,35]
	**1232**	**Amide III (proteins)** [20,34,35,44]
	1303	CH_3_/CH_2_ twisting (collagen) [34,35]
	1576	Nucleic acid modes [34,35]
	1603	C=C in-plane bending mode of phenylalanine & tyrosine [34]
	1673	Amide I (proteins) [34]

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
