# Peer review of "Discrimination of Benign and Malignant Lesions in Canine Mammary Tissue Samples Using Raman Spectroscopy: A Pilot Study"

_animals, 2020, doi:10.3390/ani10091652_

Round 1

Reviewer 1 Report

Samples from 6 female dogs which have been diagnosed to contain benign and malign tumors have been investigated using Raman spectroscopy. The Raman peaks are associated to the corresponding vibrations of biomolecules and minerals (e.g. calcium hydroxyapatite). Several fingerprint peaks have been identified e.g. the CH2 bending mode which was found to be present in the mixed carcinoma samples. The paper is well written and shows the potential and difficulties of the application Raman spectroscopy with biological tissues.

Comments in detail:

A copy of Figure 1c is lost in the upper left corner

Figures 2-6: The relation between the inset and the image at the right side remains unclear. Which is the image where the Von Kossa coloration is applied?

Page 6: “Autofluorescence maybe generated by can be produced by …” should be “Autofluorescence can be produced by …”

Author Response

The authors are grateful for the time, attention, valuable suggestions and comments, addressed below.

Reviewer #1

Samples from 6 female dogs which have been diagnosed to contain benign and malign tumors have been investigated using Raman spectroscopy. The Raman peaks are associated to the corresponding vibrations of biomolecules and minerals (e.g. calcium hydroxyapatite). Several fingerprint peaks have been identified e.g. the CH2 bending mode which was found to be present in the mixed carcinoma samples. The paper is well written and shows the potential and difficulties of the application Raman spectroscopy with biological tissues.

Comments in detail:

Reviewer: A copy of Figure 1c is lost in the upper left corner

Authors: Thank you, said figure was removed.

Reviewer: Figures 2-6: The relation between the inset and the image at the right side remains unclear. Which is the image where the Von Kossa coloration is applied?

Authors: The Von Kossa coloration was applied to samples shown in the insets while the right-side image is a microscopic one, taken from the Raman spectrometer system. For better clarification, addition information was added to Figures 2-6 as follows:

“Inset of Figure 2 is a 2.5x microscopic image of a section of sample #1 where mineralized areas are stained black by means of Von Kossa – Van Gieson modified method. A 50x microscopic image is also depicted on the right side of Figure 2, representing a mineralized area of sample #1.”

Caption of Figure 2: “Raman spectra of mineralized areas of a benign mixed tumor (sample #1) and, on the right side, the corresponding microscopic image (50x, Raman spectrometer system). Inset is the microscopic image (2.5x) of a section of sample #1 where mineralized areas are stained black by means of Von Kossa – Van Gieson modified method.”

“Figure 3 shows the spectra of the analyzed areas containing glandular secretions and cellular debris inside an ectasia duct in sample #1 and, in inset a microscopic image (2.5x) of said areas stained by means of Von Kossa – Van Gieson modified method. On the right side of Figure 3 is depicted a microscopic image (50x) of the analysed areas in sample #1containing glandular secretions and cellular debris.”

Caption of Figure 3: “Raman spectra of intraductal secretions and cellular debris in ectasia duct areas in sample #1 and, on the right side, the corresponding microscopic image (50x, Raman spectrometer system). Inset is the microscopic image (2.5x) of intraductal secretions and cellular debris stained by means of Von Kossa – Van Gieson modified method.”

“Figure 4 shows the Raman spectra of the demineralized regions in the benign mixed tumor and, in inset a microscopic image (2.5x) of said areas stained by means of Von Kossa – Van Gieson modified method. On the right side of Figure 4 is depicted a microscopic image (50x) of a demineralized region in sample #1.”

Caption of Figure 4: “Raman spectra of demineralized areas of sample #1 and, on the right side, the corresponding microscopic image (50x, Raman spectrometer system). Inset is the microscopic image (2.5x) of demineralized areas stained by means of Von Kossa – Van Gieson modified method.”

Caption of Figure 5: “Raman spectra of mineralized areas of bone metaplasia of the mixed carcinoma (sample #3) and, on the right side, the corresponding microscopic image (50x, Raman spectrometer system). Inset is the microscopic image (2.5x) of a section of sample #3 where mineralized areas are stained black by means of Von Kossa – Van Gieson modified method.”

Caption of Figure 6: “Raman spectra of demineralized areas of the mixed carcinoma (sample #3) and, on the right side, the corresponding microscopic image (50x, Raman spectrometer system). Inset is the microscopic image (2.5x) of demineralized areas stained by means of Von Kossa – Van Gieson modified method.”

Reviewer: Page 6: “Autofluorescence maybe generated by can be produced by …” should be “Autofluorescence can be produced by …”

Authors: Thank you, the sentence was corrected

Reviewer 2 Report

The manuscript “Discrimination  of  benign  and  malignant  lesions  in canine  mammary  tissue  samples  using  Raman Spectroscopy: a pilot study” by Diana Dantas, Liliana Soares, Susana Novais, Rui Vilarinho, J. Agostinho Moreira, Susana Silva, Orlando Frazão, Teresa Oliveira, Nuno Leal, Pedro Faísca, Joana Reis shows the possibility of using Ranam spectroscopy for detection of  breast cancer on the example of biological material from dogs. Mostly, the conclusion could be done by some pikes in spectra arising. Very minor changes should be done.

The paper has good structure and easy to understand for any kind of readers. In general, there are enough information and analytical discussions, but:

  1. I’ll kindly ask to add some info in introduction section to concentrate more attention for Raman techniques for cancer detection (to share world experience).
  2. Please, indicate the minimum number of spectra for each sample to obtain a reliable result.

Author Response

The authors are grateful for the time, attention and valuable suggestions and comments, addressed below.

Reviewer #2

The manuscript “Discrimination of benign and malignant lesions in canine mammary tissue samples using Raman Spectroscopy: a pilot study” by Diana Dantas, Liliana Soares, Susana Novais, Rui Vilarinho, J. Agostinho Moreira, Susana Silva, Orlando Frazão, Teresa Oliveira, Nuno Leal, Pedro Faísca, Joana Reis shows the possibility of using Ranam spectroscopy for detection of breast cancer on the example of biological material from dogs. Mostly, the conclusion could be done by some pikes in spectra arising. Very minor changes should be done.

The paper has good structure and easy to understand for any kind of readers. In general, there are enough information and analytical discussions, but:

  1. I’ll kindly ask to add some info in introduction section to concentrate more attention for Raman techniques for cancer detection (to share world experience).

Authors: Thank you, introduction was modified as suggested. Additional information was added as follows:

In recent decades, Raman spectroscopy has been under considerable attention concerning clinical oncology applications, for ex vivo studies and during surgery of neoplasms affecting the brain, breast, lung, skin, uterus, and GI tract [14]–[17]. In fact, our research group has successfully shown the feasibility of Raman spectroscopy for discriminating between adenocarcinomatous and normal mucosal formalin-fixed colonic tissues [18].

(…)

To the authors’ knowledge Raman has also been used previously in canine mammary neoplasms by Birtoiu et al. (2015) [6] and in two clinical cases as described by Munteanu et al (2017) [23]. Both works used ex vivo, freshly excised, unprocessed samples, and a 632 nm wavelength laser. Birtoiu et al. refer 10 samples from female cats and dogs, without specification e the histopathological diagnosis is not described; Munteanu et al. refer to a case of solid carcinoma and another with lobular hyperplasia. The present study aimed to explore the use of Raman spectroscopy to characterize, in histology sections, the lesions associated with different neoplasms in canine mammary tissue, with a focus in those with mineralization, and explore its potential to discriminate between benign and malignant lesions in this species, systematically identifying characteristic fingerprints, through a comparative oncology approach.

  1. Please, indicate the minimum number of spectra for each sample to obtain a reliable result.

Authors: Thank you for this very interesting question. We believe there is no consensus on this subject. It depends on the homogeneity of sample itself and on the consistency of the spectra. The authors chose to acquire a minimum of 3 spectra per site and use a higher number of sites considering the expected heterogeneity of canine mammary tumors. We added to the text “Considering the consistency of spectra at each given point and the heterogeneous nature of canine mammary tumors, a minimum of 3 spectra per site was acquired, from at least 15 sites per sample.”

Reviewer 3 Report

In this manuscript, Dantas et al have discussed how Raman spectrometry can be used to differentiate between non-neoplastic and neoplastic mammary gland lesions and benign lesions from malignant lesions in dogs. Although this provide some useful insight on how this technique could be useful in prognostic point of view, the manuscript have some major issues to be addressed.

  1. Authors need to be clear whether this technique has been performed for canine mammary neoplasms previously and if so how the present findings are novel or different from the previous findings.
  2. The study has used relatively small number of samples. It is better to increase the sample number if possible. Otherwise it should be mentioned as a limitation of the study in the discussion. Further, only a few histological sub-types have been used in the study. Canine mammary neoplasms have a wide histological diversity than human breast cancers and therefore authors need to clarify on what basis they have selected these particular sub-types for this study. Additionally, it is not clear why the adjacent non-neoplastic mammary tissue in neoplasms have been assessed only in a single tumour and not in others. It is better if the authors could examined the adjacent non-neoplastic tissues also for all the included neoplasms.
  3. The results and discussion section is too lengthy and bit difficult to read. Authors may need to revise the results and discussion to be more precise and concise. 
  4. Although it is discussed in the introduction about the sensitivity and specificity of Raman spectrometry, those have not been evaluated in this study. If sensitivity and specificity of the technique have not been analysed in this study, authors need to explain why it could not be accomplished. 
  5. The manuscript is not clear about how Raman spectrometry is better than other prognostic analysis methods and it is necessary to briefly discuss it in introduction or in discussion. 

Other minor points include

Summary

  1. "Dogs spontaneously suffer from tumors that mirror carcinogenic conditions similar to those found in humans" It may be not appropriate to use the term "mirror" as the mammary lesions in humans and dogs are not identical

Introduction

  1. It is not clear in the introduction whether Raman spectrometry has been previously performed in canine mammary neoplasms or not. The authors need to be clear about this and if it has been performed it is necessary to briefly describe how the present findings/approach is different from that.
  2. "Tumors may be classified as benign or malignant" Authors need to be clear whether they refers to benign or malignant status of a neoplasm based on histology.
  3. "However, spontaneous mammary tumors are common in dogs and cats. About 42% of all tumors in the female dogs and 85% in cats affect the mammary gland" The authors need to re-check the statement about 85% of cat neoplasms to be mammary gland tumours. It should be less than that. 

Material and methods

  1. "decalcified with RDO" It is necessary to include the extended form of this abbreviation.
  2. "sections were previously deparaffinized" It is not clear what is meant by "previously deparafinized". 
  3. Raman spectroscopy was performed using a Renishaw InViaTM Qontor® Confocal spectrometer,
    equipped with lasers of different wavelengths of 532, 633 and 785 nm, respectively" The authors need mention why these specific wave lengths have been used.

Author Response

Reviewer #3

In this manuscript, Dantas et al have discussed how Raman spectrometry can be used to differentiate between nonneoplastic and neoplastic mammary gland lesions and benign lesions from malignant lesions in dogs. Although this provide some useful insight on how this technique could be useful in prognostic point of view, the manuscript have some major issues to be addressed.

  1. Authors need to be clear whether this technique has been performed for canine mammary neoplasms previously and if so how the present findings are novel or different from the previous findings.

Authors: The following was added to Introduction:

To the authors’ knowledge Raman has also been used previously in canine mammary neoplasms by Birtoiu et al. (2015) and in two clinical cases as described by Munteanu et al (2017). Both works used ex vivo, freshly excised, unprocessed samples, and a 632 nm wavelength laser. Birtoiu et al. refer 10 samples from female cats and dogs, without specification and the histopathological diagnosis is not described; Munteanu et al refer to a case of solid carcinoma and another with lobular hyperplasia.

The present study aimed to use Raman to characterize, in histology sections, the lesions associated with different neoplasms in canine mammary tissue, including those with mineralization, and explore its potential to discriminate between benign and malignant lesions in this species, systematically identifying characteristic fingerprints, through a comparative oncology approach.”

In “results and discussion” the following was added:

 “Only one of two previous studies published in Raman application to canine mammary neoplasms refers the histopathological diagnosis of two clinical cases, both different from the ones included in the present study and none with mineralization. These studies were performed in freshly excised tissue, unlike our samples, and Raman spectra were acquired with 632 nm laser, unlike the present study.”

  1. The study has used relatively small number of samples. It is better to increase the sample number if possible. Otherwise it should be mentioned as a limitation of the study in the discussion. Further, only a few histological sub-types have been used in the study. Canine mammary neoplasms have a wide histological diversity than human breast cancers and therefore authors need to clarify on what basis they have selected these particular sub-types for this study. Additionally, it is not clear why the adjacent non-neoplastic mammary tissue in neoplasms have been assessed only in a single tumour and not in others. It is better if the authors could examined the adjacent non-neoplastic tissues also for all the included neoplasms.

Authors: As explained in section 2.1 the biopsies “were taken from areas with increased radiopacity” It was not a goal of the study to specifically address adjacent non-neoplastic tissues. Samples were selected based on suspicion of mineralization and being a pilot, exploratory study, it was not aimed at extensive characterization of all canine mammary tumors, as the authors are well aware, they are very heterogeneous. The following sentence was added to Introduction “The present study aimed to explore the use of Raman spectroscopy to characterize, in histology sections, the lesions associated with different neoplasms in canine mammary tissue, with a focus in those with mineralization, and explore its potential to discriminate between benign and malignant lesions in this species, systematically identifying characteristic fingerprints, through a comparative oncology approach.” In face of the obtained results, the research team is now determined to extend the application of Raman of a larger number of samples.

In “results and discussion”, the following sentence was modified to more clearly address why authors believe a larger number of samples will be included in future work.

“However, challenges are also posed by the heterogeneous nature of canine mammary tumors that makes laser focusing in a uniform cell population unlikely. The existence of strong laser-induced fluorescence in the non-neoplastic samples included initially also limits comparison with non-neoplastic tissues, with the present methodology of sample preparation. The research team is currently working in solutions to circumvent autofluorescence, allowing acquisition from control tissues, and applying the methodology to a larger number of samples, better representative of the diversity of canine mammary tumours, with subsequent statistical analysis.”

  1. The results and discussion section is too lengthy and bit difficult to read. Authors may need to revise the results and discussion to be more precise and concise.

Authors: This being a multidisciplinary study, the authors aimed at including information relevant for the different scientific fields, from Physics to Biochemistry and Comparative Oncology.

In the section of “results and discussion” four subsections were added for better understanding the analyzed results. Additional information was also added as follows:

“The following sections 3.1 and 3.2 present the Raman spectroscopy results of samples #1 and #3, respectively, and the analysis of major assignments obtained with the proposed technique, namely the ones presented in Table 3. Due to similar results achieved from the analysis of Raman spectra of samples #1 and #2, apart from being two mixed-type benign tumors with bone metaplasia, only results from Sample #1 are shown.

Section 3.3 briefly discusses the major Raman assignments achieved from the analysis of sample #4 – a malignant tumor. Although not having bone metaplasia, it presents common Raman peaks with samples #1 to #3, as they all share tissue changes related with neoplasia.

Section 3.4 summarizes the main Raman fingerprints achieved for all samples under study (#1 to #4) and its major assignments.”

3.4. Main Raman Fingerprints:

“As previously discussed, the band at 1673 cm-1 is assigned to the amide I mode originating mainly from the polypeptide backbone and protein secondary structure and it is present in all samples (#1 to #4). This Raman band has been identified as indicative of tissue changes associated with disease, namely neoplasia [20, 34, 35].

It is also worth notice that all samples (#1 to #4) have in common the Raman band at 1454 cm-1, assigned to CH3 and CH2 deformation (proteins), an indicator of unhealthy tissues [34]. Tissue samples #1 and #2 also have in common the band at 1452 cm-1, which is assigned to the C-H bending mode and it has been associated with lipids and cardiac valvular calcification [27,37]; 1453 cm-1 has been associated with tumor structural modes of proteins [34].”

  1. Although it is discussed in the introduction about the sensitivity and specificity of Raman spectrometry, those have not been evaluated in this study. If sensitivity and specificity of the technique have not been analysed in this study, authors need to explain why it could not be accomplished.

Authors: The submitted version of manuscript animals-906744 does not discuss sensitivity and specificity of Raman spectrometry in introduction. The present study is exploratory and focused in canine neoplasms for which there was no Raman spectroscopy data available until now and that presented suspicion of mineralization on imagiology. The approach is comparative, considering that this technique has been more widely applied to human breast cancer. To be able to conclude on its sensitivity and specificity, a much larger number of samples would be needed, and a different design. Also, as discussed in our document, the research team is currently working in solutions to circumvent autofluorescence, an important limitation in spectra acquisition with the present methodology.

  1. The manuscript is not clear about how Raman spectrometry is better than other prognostic analysis methods and it is necessary to briefly discuss it in introduction or in discussion.

Authors: It is not the purpose of this study to show that Raman spectroscopy is an alternative prognostic analysis method. Raman spectroscopy is highly attractive as a complementary technique for other clinical applications and in diagnosis, as mentioned in conclusions. This issue was clarified in introduction - please see answer to question 1.

Other minor points include:

Summary

  1. "Dogs spontaneously suffer from tumors that mirror carcinogenic conditions similar to those found in humans"

It may be not appropriate to use the term "mirror" as the mammary lesions in humans and dogs are not

Identical

Authors: Although the sentence "Dogs spontaneously suffer from tumors that mirror carcinogenic conditions similar to those found in humans" refers to neoplastic disease in general, we understand the reviewer’s concern. For clarity, the sentence was changed to “Canine neoplastic disease occurs spontaneously and has comparable clinical presentation and pathophysiology to corresponding human cancers.”

Introduction

  1. It is not clear in the introduction whether Raman spectrometry has been previously performed in caninemammary neoplasms or not. The authors need to be clear about this and if it has been performed it is necessary to briefly describe how the present findings/approach is different from that.

Authors: Please see answer to question 1.

  1. "Tumors may be classified as benign or malignant" Authors need to be clear whether they refers to benign or malignant status of a neoplasm based on histology.

Authors: Sentence now reads as “Tumors may be classified as benign or malignant based on histopathology”

  1. "However, spontaneous mammary tumors are common in dogs and cats. About 42% of all tumors in the female dogs and 85% in cats affect the mammary gland" The authors need to re-check the statement about 85% of cat neoplasms to be mammary gland tumours. It should be less than that.

Authors: Sentence was corrected to “About 42% of all tumors in the female dogs and 17% in cats affect the mammary gland”, according to reference: V. Zappulli, R. Rasotto, D. Caliari, M. Mainenti, L. Peña, M. Goldschmidt, and M. Kiupel, “Prognostic evaluation of feline mammary carcinomas: a review of the literature,” Veterinary pathology, vol. 52, no. 1, pp. 46–60, 2015.

Material and methods

  1. "decalcified with RDO" It is necessary to include the extended form of this abbreviation.

Authors:  Corrected to RDO® Rapid Decalcifier

  1. "sections were previously deparaffinized" It is not clear what is meant by "previously deparafinized".

Authors: Sentence corrected to “routinely deparaffinized and hydrated”, to avoid lengthy description of routine steps of washes in xylene and decreasing concentration ethanol solutions.

  1. Raman spectroscopy was performed using a Renishaw InViaTM Qontor® Confocal spectrometer, equipped with lasers of different wavelengths of 532, 633 and 785 nm, respectively" The authors need mention why these specific wave lengths have been used.

Authors: The wavelength used, although mentioned in the text (“The 532 nm laser line was used for excitation.”), was modified as follows and justified in Results and Discussion:

“2.2. Raman Spectroscopic Measurements:

The 532 nm laser line was chosen for excitation of all samples under analysis as the remaining wavelengths provided strong background fluorescence. (…) Each Raman spectrum was acquired with an exposure time of 10 sec.”

  1. Results and Discussion:

“All tissue samples were previously tested by the three lasers available in the Raman spectrometer system, i.e. emitting at the wavelengths of 532, 633 and 785 nm. The Raman spectra acquired with 633 and 785 nm laser lines shown strong background fluorescence, inhibiting the analysis of the acquired data. The Raman spectra acquired with the 532 nm laser presented much lower fluorescence and, thus, it was chosen for the excitation of all samples for further analysis, an approach applied by other authors [16,21,22,26–28].”

Round 2

Reviewer 3 Report

I have gone through the rebuttal and the revised manuscript forwarded and the changes seems appropriate. Therefore, I confirm that the revision process is completed.

This manuscript is a resubmission of an earlier submission. The following is a list of the peer review reports and author responses from that submission.

Round 1

Reviewer 1 Report

Although the authors explore a topic of great relevance in veterinary oncology, and apply a technique that could be promising as an auxiliary diagnostic tool in canine mammary tumors, this investigation presents major conceptual flaws.

The text is not well written, sometimes is poorly structured, especially regarding the subject of veterinary pathology and oncology, with many naive statements and little precision in the terminology employed. The intervention of someone with experience in the field of veterinary oncology is lacking, which is essential in a study of this nature. Indeed, there is a big difference in the rigor with which M&M 2.1 and 2.2 / 2.3 are described.

Chondroid/osseous metaplasia is a common feature in canine mammary tumors. However, in most cases, the malignant neoplastic component is restricted to the epithelial population, not involving the mesenchymal one; i.e., the cartilage and bone tissue does not exhibit malignancy criteria - these are called malignant mixed tumors. In this sense, it is essential to understand whether the chondroid / bone component evaluated has histological criteria of malignancy or not. In this study it is not known whether the bone component evaluated is benign in lesion A (benign mixed tumor) and in lesion B (malignant mixed tumor), or if the lesion B is a carcinosarcoma, in which the mesenchymal component (bone, in this case) is malignant.

Furthermore, it is not conceivable to state that this technique allows differentiation between benign and malignant neoplasms using only one sample of benign tumor and one sample of a malignant one. Besides, it is impossible to characterize the canine mammary tumors (and adjacent mammary tissue) profile using only 2 samples.

On the basis of the exposed I do not recommend the paper for publication in a magazine with the prestige and high impact of Animals.

Author Response

The authors would like to thank the reviewer’s valuable comments, which we feel helped to improve the manuscript.
English language and style have been reviewed.
Major revisions were made, namely concerning sample characterization. Special care was taken in improving section 2.1, both in detail and language. One table and a figure were inserted in this section, the first characterizing all the samples studied in this exploratory work and the second depicting the histopathology findings. A total of seven samples from six animals were used in this study but strong autofluorescence impaired reading in three of the samples.
Both previous samples A and B (now #1 and #3) were, in fact, of mixed-type tumors. We have now added spectral data from an additional benign mixed tumor (#2) and a carcinosarcoma (#4). Table 2 now shows the number of sites and the number of spectra per site obtained in each sample. We also added table 9, summarizing the main differentiating peaks in between the samples, including those already signaled as fingerprints for malignancy in neoplastic disease.
Raman spectroscopy is a sensitive tool for assessing the molecular composition and structure of samples. In the case of neoplastic diseases, the known histopathologic changes translate into predominance of protein, nucleic acid modes and certain vibrational models that consistently are referred in the literature. These findings make sense when considering the changes at the tissue and cellular level in neoplastic diseases, and criteria to distinguish benign lesions from malignant ones. In the present work we identify which of these characteristic modes were consistently present in similar regions of the same sample.

Reviewer 2 Report

In this study, authors use a special technique, the Raman Spectroscopy, to characterize canine mammary tissue with neoplastic lesions, especially microcalcifications. This technique is widely used in human medicin to diagnose and evaluate breast cancer in women. Authors found that by identifying peak bands, the existence of a tumor can be proven. Furthermore that recognition of structural protein changes might enable differentiation between malignant and benign tissue alterations.

In general, this study is highly interesting. It in fact emphasizes the necessity of a closer cooperation between different medical disciplins, and between human and veterinary medicine. Unfortunately only two samples of mammary tumors were taken, the study therefore is a pilot study, which is addressed in the conclusions. Even though at this point the diagnostic value is limited, the technique is highly interesting for research purposes, especially for the development of biomarkers.

Page 2, 5th paragraph, last sentence: sentence is too long. Proposal: ….mammary tumors. These facts validate this recognized spontaneous animal model …

The microscopic images could be better described (magnification, bars)

Page 8, second paragraph under table 5: Through the analysis of these spectra

Page 9, first paragraph under table 6…however,..

Author Response

The authors would like to thank the reviewers' valuable comments. We tried to revise the whole manuscript accordingly.

We have made a major revision of the manuscript and highlighted in the title the fact that it is still a preliminary study. Overall, we have addressed seven samples from six animals.

We have added information on all samples, both benign and malignant, as summarized in table 1 in section 2.1, latter discussed in the Results and discussion section.

We have also included in section 2.1, in Figure 1, images of the histopathology findings in all the samples, that include a scale. In the microscopic von Kossa images, we have added amplification.

We have reviewed and corrected the English language.

Reviewer 3 Report

This is a very interesting study, with highly promizing applications in the treatment of breast cancer in dogs. Unfortunately, the presentation of the research, does not help the readers in their attempt to evaluate the process and the results. There is a continuous presence of information that should be in the Material and Methods section, into the Results section. There is a confusing way to present the protocol of the study, with an absence of the usual steps of presentation of the Material and Methods (we would appreciate to have more details about the origin of the samples, the dogs, their age, ...), everything about the analysis of the samples should be in this section, not in the results. Since the paper is raising the question of the accuracy of Raman spectroscopy, we would like to know how the authors have organized to control this accuracy. Is there any statistical analysis, correlation? that is checked. If yes, which method did they use?

This way to write the paper is making it very confusing, and this is a pity with such an interesting work.

Author Response

The authors would like to thank the reviewer’s valuable comments, which the authors feel helped improving the manuscript.

In this manuscript with major revisions, we have added information on the samples and we have added more samples, both benign and malignant, as summarized in table 1, in section 2.1.We have also included in section 2.1, in Figure 1, images of the histopathology findings in all the samples.

We also added additional information on the number of sites and spectra readings in all samples, as summarized now in Table 2, in section 2.2.

We have, therefore, aimed at reinforcing the Material and Methods section, and detailed and discussed the results from the 4 samples from which good quality spectra were obtained. We have tried to simplify the language and results presentation and discussion.

Findings were consistent in similar areas of a same sample, and when spectra were heterogenous as it happened in sample #4, this is mentioned and discussed.

We have added Table 9 i Results section, that summarizes the peaks found that differentiated samples, including those that are reported in the literature as associated with neoplastic tissue.

We refer why we did not perform statistical analysis in this exploratory study, as we were aiming at exploring the presence of known fingerprints of neoplastic changes and tumor aggressiveness, as previously reported in the literature for humans.

In face of these promising results, the authors now hope  to enlarge the number of samples, adapting the methodology to overcome difficulties associated with sample processing in histology and laser-induced fluorescence.

We have reviewed and corrected the English language